# Inertial Propulsion of a Mobile Platform Driven by Two Eccentric Bodies

Stelica Timofte [1], Calin-Octavian Miclosina [2], Vasile Cojocaru [2], Attila Gerocs [1,3] and Zoltan-Iosif Korka [1,2,*]

1   Doctoral School of Engineering, Babes-Bolyai University, University Center Resita, 320085 Reșița, Romania; stelica.timofte@ubbcluj.ro (S.T.); atti.gerocs@gmail.com (A.G.)
2   Department of Engineering Sciences, Babes-Bolyai University, University Center Resita, 320085 Reșița, Romania; calin.miclosina@ubbcluj.ro (C.-O.M.); vasile.cojocaru@ubbcluj.ro (V.C.)
3   Automation, Industrial Engineering, Textiles and Transportation Department, "Aurel Vlaicu" University of Arad, 310032 Arad, Romania
*   Correspondence: zoltan.korka@ubbcluj.ro; Tel.: +40-745-911-887

**Abstract:** In recent decades, enthusiasts and researchers from all over the world have invested a lot of energy and time in trying to develop linear propulsion systems using the inertial force generated by two masses revolving in opposite directions. The authors present an original concept for the propulsion system of a mobile platform by using the inertial force generated by two eccentric bodies that are revolving in opposite directions. Based on this approach, a wheeled vehicle was designed, built using additive manufacturing techniques, and finally tested. Validation of the concept and verification of the analytically derived kinematic parameters were carried out by 3D modeling and motion simulation performed in SolidWorks 2023. It was concluded that the initial position of the eccentric bodies determines the direction of movement. Moreover, a favorable correlation of the eccentric bodies' masses with the entire platform mass is able to ensure the movement at an oscillating but permanently positive speed.

**Keywords:** eccentric bodies; inertial propulsion; mobile platform

## 1. Introduction

The concept of inertial propulsion using eccentric masses in order to produce linear motion in a body to which they are attached is an older topic [1], dating back to the time when ancient Greek athletes used weights to lengthen their long jumps [2] (p. 153). Nevertheless, it is a topic that is met with great interest even nowadays [3–6]. Their advantage over classic engine propulsion systems consists in the fact that they allow the elimination of the transmission system to the wheels, making them a viable alternative for driving boats or submarines when the propeller and its sealing system can be eliminated. In addition, they could be suitable for driving vehicles on very slippery surfaces (mud or ice), where no tires would provide the grip necessary for movement.

The concept of contra-rotating masses could be used to propel vehicles on land, in water, and in space. However, from a strictly mechanical point of view, the following controversy exists: since inertial forces are internal forces, the rotation of the masses does not produce an external force (thrust) that would accelerate the center of mass of the system. Nevertheless, many scientists [7–9] have made an effort to address this contentious issue, and even NASA has carried out a six-year research program in this area [10], but providing only qualitative information.

One of the most studied [11–13] inertial propulsion systems is Dean's drive, also known as Dean's apparatus. It was patented in 1959 [14] and uses two contra-rotating eccentric masses mounted on a carrier, which is guided on vertical rails. Tension springs are used to attach the carrier to the housing.



Many inertial drives exist, but so far only as patents. One such example is the propulsion system suggested by Boden [15]. It produces a unidirectional thrust by converting the energy of two identical assemblies of weights that are spinning in opposite directions. The peculiarity of the system consists of its rotating elements, which are moving with variable gyration radii around a fixed point in order to achieve an imbalanced centrifugal force. As a result, a propulsion force in a given direction is generated.

The propulsion apparatus suggested by Cuff [16] includes a device for varying the rotation radii of several masses and for proper direction adaptation of the resultant unbalanced force produced by these rotating weights. The device includes two fixed circular cams installed eccentrically. The cam followers are joined with rods, which are successively connected to the gyrating masses. The direction of the resultant unbalanced force can be modified by the in-concordance rotation of the commonly fixed cams.

Similarly, Dobos [17] proposes an apparatus comprising a platform mounted perpendicularly on a shaft. The shaft also supports a disk on which several reservoirs containing liquid are placed around the circumference edge. A hollow piston with a piston rod extending outward from the reservoir is buoyantly positioned in each reservoir. An adjustable moving cam that has a cam track eccentrically positioned with regard to the shaft assembly's axis is mounted onto the shaft assembly itself. For moving into and out of contact with the cam track, the piston rod end sections are placed opposite the cam track. Advancing the cam to a predetermined position displaces the pistons in the reservoirs as the disk rotates the cam. This is accomplished by a load head that imparts a downward force against a bent but flexible shaft, which in turn engages a shoulder on the cam's maximum radial sector. The relative displacement of the liquid in the reservoirs by the movement of the pistons in response to the contact of the piston rods with the cam track creates an unbalanced centrifugal force that propels the platform in a preselected linear direction.

Later on, Deschamplain [18] patented a propulsion device using fluids to convert rotary motion into linear movement. His system includes a centrifuge with a cell that is partly filled with water. While the centrifuge is being driven by a motor, the centrifugal force forces the fluid to move outward. The assembly also contains an object that is positioned at its exterior end in water and at its interior end in air. A rod that rotates with the same speed as the centrifuge is placed parallel to the axis of the centrifuge at a certain distance and is connected to the object by a coupling. The centrifugal forces generated by the synchronous rotation of the centrifuge and the rods are used to generate linear motion.

In the same year, Murray [19] obtained a patent for a mechanical force generator that converts the cinematic energy of centrifugal force into propulsion force using a cage assembly that rotates around its longitudinal axis. The cage rotates secondary shafts, which turn sets of eccentrics to generate a net force in a direction that is transverse to the rotation axis of the cage assembly. Two pairs of eccentrics turn in such a manner that for each 90° rotation of the carrier cage, the pairs of eccentrics have their centers of mass placed rather between a balanced and an unbalanced status, but, at each 90° rotation angle, one pair of eccentrics is always producing a power stroke.

More recently, Loukanov [20] presented the concept of a unidirectional mobile robot, by using two eccentric masses rotating synchronously. The motion is obtained by both inertial and frictional forces. In order to avoid resonance, the mobile platform includes a spring and damper system. Furthermore, one-way roller bearings mounted in the hubs of the wheels ensure the unidirectional displacement of the vehicle, suppressing the unwanted backward movement. Based on this concept, the author built a wheeled vehicle and tested it on surfaces with varying degrees of friction and supporting different weights.

However, we note that none of the above-mentioned propulsion drive concepts have been practically materialized, or at least remain unreported in the scientific literature. Therefore, it remains unclear to what degree inertial propulsion systems work in the real world, if at all.

This paper presents an original concept for the propulsion system of a mobile platform by using the inertial force generated by two eccentric masses that are rotating in

opposite directions. The aim of this research was to prove the validity of the new concept through numerical simulation and experimental testing without analyzing aspects related to efficiency and power consumption. These aspects will be the subject of further research.

By contrast to the similar concepts that were presented above, the proposed inertial propulsion system (IPS) ensures a movement with an oscillating, but permanently positive speed. Moreover, the displacement speed may be favorably set by a proper correlation of the eccentric bodies' masses with the entire platform mass. Furthermore, the initial position of the eccentric bodies determines the direction of movement. Based on this approach, a wheeled vehicle was designed and built using 3D printing techniques. Validation of the concept and verification of the analytically derived kinematic parameters were accomplished by 3D modeling and motion simulation performed in SolidWorks. Finally, the system was experimentally tested.

## 2. Description of the Inertial Propulsion System

The IPS developed by the authors consists of two identical eccentric masses, 6/1 and 6/2, which are driven synchronously, in opposite directions of rotation, by the e-motor 1, which is operable in the speed range 0–3000 rot/min.

The drive chain of the eccentrics includes three bevel wheels 4 and two identically multiplying cylindrical gear pairs consisting of the wheels 6 and the pinions 7.

The central bevel gear 4/1 is driven by the shaft 2, which is supported by a pair of ball bearings mounted in the bearing block 3. The cylindrical gears and the eccentrics are also mounted on shafts supported by pairs of ball bearings assembled in the housing 8. Both the bearing block 3 and housing 8 are fastened on the wooden plate 10, which is supported by four rubber wheels 11, similarly to a previously presented IPS [3]. The supports 9 and some hexagonal nuts may adjust the correct engagement of the bevel gears.

Except for the shafts, the eccentric masses, the bearings, and the housing supports, all the other components shown in Figure 1 were obtained by additive manufacturing from Polylactic Acid (PLA) type filament printed by Fused Deposition Modeling (FDM) technology.

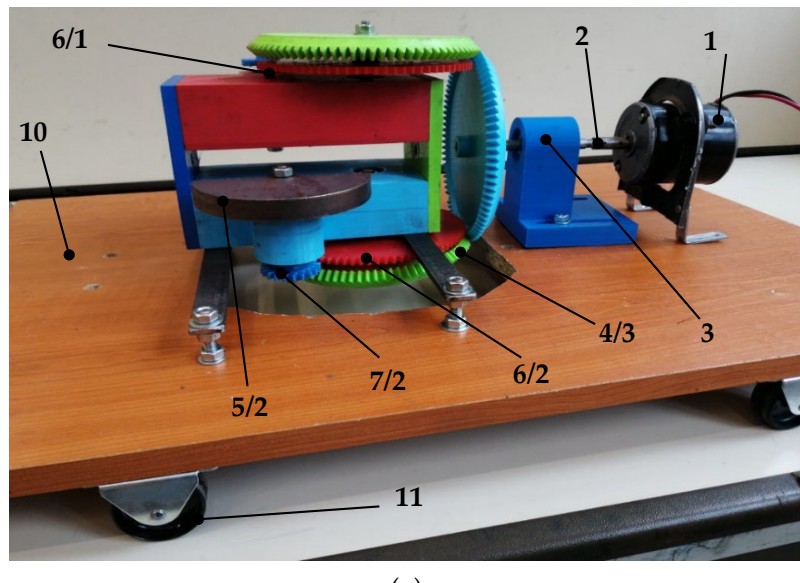

(a)

**Figure 1.** *Cont.*

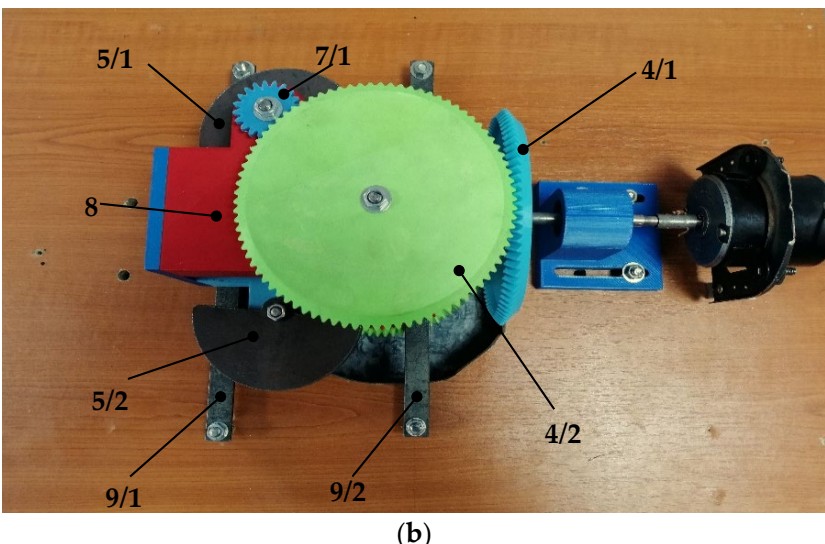

**(b)**

**Figure 1.** Inertial propulsion drive with counter-rotating eccentrics. (**a**) Front view; (**b**) upper view.

### 3. Analytical Investigation of the Inertial Propulsion System

For the deduction of the motion equations for the IPS, we considered two eccentric bodies, which are each materialized by a mass $m$ connected to a rigid rod of length $r$. The two rotating masses are attached to the mobile platform of mass $M$ and are rotating in a synchronized manner, in opposite directions. As the masses are contra-rotating perfectly synchronously, the potential movement will take place only along one axis ($x$ in Figure 2), while along the other axis ($y$), the potential movements nullify each other.

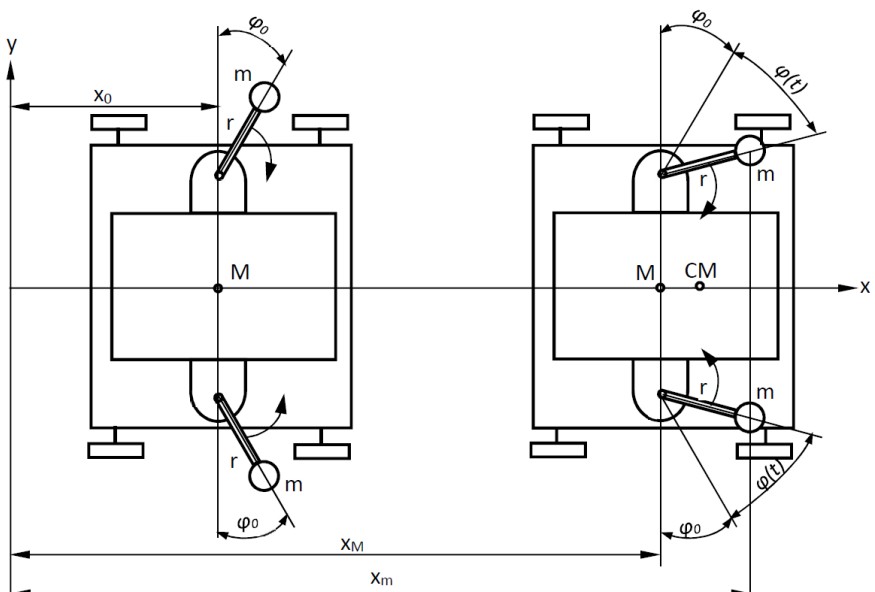

**Figure 2.** Initial and arbitrary positions of the IPS.

In relation to the fixed coordinate system $xOy$, at the initial moment ($t = 0$), the platform's center of mass (COM) is located at the distance $x_0$, while the rods form an angle $\varphi_0$ with the $y$-axis. At the arbitrary time $t > 0$, the system will travel a distance along the $x$ axis, so that the momentary coordinate of the platform COM will be $x_M$. The abscissa of the masses $m$ is $x_m$ and the angle described by the rotating masses is denoted by $\varphi(t)$.

The inertial force generated by the so-called "driving masses" $m$ will act only along the $x$ axis, directed opposite to the movement of the platform, and will be equal to $-2m\ddot{x}_m$. Thus, Newton's second law can be written for the platform $M$ as follows:

$$M\ddot{x}_M = -2m\ddot{x}_m. \tag{1}$$

Given that:

$$x_m = x_M + r\sin[\varphi(t) + \varphi_0], \tag{2}$$

and $\varphi(t) = \omega \cdot t$, where $\omega$ is the angular velocity of the eccentric bodies, the first derivative of $x_m$ may be expressed as:

$$\dot{x}_m = \dot{x}_M + \omega r\cos(\omega t + \varphi_0), \tag{3}$$

or:

$$v_m(t) = v_M(t) + \omega r\cos(\omega t + \varphi_0), \tag{4}$$

which means that the velocity of each driving mass is a sum of an absolute motion (motion of the platform $v_M$) and a relative motion.

Further, the second derivative of $x_m$ is:

$$\ddot{x}_m = \ddot{x}_M - \omega^2 r\sin(\omega t + \varphi_0). \tag{5}$$

Therefore, Equation (1) becomes:

$$M\ddot{x}_M = -2m\ddot{x}_M + 2m\omega^2 r\sin(\omega t + \varphi_0), \tag{6}$$

or:

$$\ddot{x}_M(M + 2m) = 2m\omega^2 r\sin(\omega t + \varphi_0), \tag{7}$$

and the acceleration of the platform is:

$$\ddot{x}_M = \frac{2m\omega^2 r}{M + 2m}\sin(\omega t + \varphi_0). \tag{8}$$

The platform's velocity is obtained by time integration of Equation (8):

$$v_M = \dot{x}_M = \int_0^t \ddot{x}_M dt = \int_0^t \frac{2m\omega^2 r}{M + 2m}\sin(\omega t + \varphi_0) = \frac{2m\omega^2 r}{M + 2m} \cdot \left[-\frac{\cos(\omega t + \varphi_0)}{\omega} + C\right], \tag{9}$$

where $C$ is an integration constant. Regrouping the terms:

$$v_M = \frac{2m\omega^2 r}{M + 2m}C - \frac{2m\omega r}{M + 2m}\cos(\omega t + \varphi_0) \tag{10}$$

At start ($t = 0$) the velocity of the platform is:

$$v_{M,0} = \frac{2m\omega^2 r}{M + 2m}C - \frac{2m\omega r}{M + 2m}\cos\varphi_0, \tag{11}$$

Further, considering the whole system of masses ($M + 2m$) with the center of mass located in *CM*, the law of impulse conservation can be written as:

$$(M + 2m)v_{CM} = 2mv_m + Mv_M, \tag{12}$$

or:

$$v_{CM} = \frac{2mv_m + Mv_M}{M + 2m}. \tag{13}$$

Applying Equation (4), Equation (12) becomes:

$$v_{CM} = \frac{2m}{M + 2m}[v_M + \omega r\cos(\omega t + \varphi_0)] + \frac{M}{M + 2m}v_M, \tag{14}$$

or:

$$v_{CM} = v_M + \frac{2m\omega r}{M + 2m}\cos(\omega t + \varphi_0). \tag{15}$$

and:

$$v_M = v_{CM} - \frac{2m\omega r}{M + 2m} cos(\omega t + \varphi_0) \,. \tag{16}$$

For the initial condition ($t = 0$), Equation (15) becomes:

$$v_{M,0} = v_{CM,0} - \frac{2m\omega r}{M + 2m} cos\varphi_0 \,. \tag{17}$$

Evaluating the Equations (10) and (16), it can be concluded that, at start ($t = 0$), the initial velocity of the whole system's *CM* is:

$$v_{CM,0} = \frac{2m\omega^2 r}{M + 2m} C, \tag{18}$$

meaning that the systems *CM* will move forward only if the constant $C \neq 0$.

Furthermore, analyzing Equation (18), one can conclude that the initial speed of the vehicles center of mass ($v_{CM,0}$) is the higher, the more:

- The platform's mass *M* is smaller, compared to the driving masses *m*;
- The driving masses are bigger;
- The rotational speed of the driving masses is higher;
- The length of the rods (*r*) on which the masses are attached is higher.

Based on Equation (18), the platforms velocity expressed in Equation (10) becomes:

$$v_M = v_{CM,0} - \frac{v_{CM,0}}{\omega C} cos(\omega t + \varphi_0) \tag{19}$$

which means that the velocity ($v_M$) consist of a linear component ($v_{CM,0}$) and an oscillating one.

### 4. Motion Simulation of the Vehicle

For this purpose, we have exploited the capabilities of the Motion module, with Motion Analysis (MA) option capabilities, from SolidWorks (SW). MA was developed to simulate and analyze the motion of an assembly with high accuracy while incorporating external effects such as forces, friction, dampers, or springs. In MA, Newton's second law of motion and Euler's second law of motion govern the movement of a rigid body.

For solving the equation of motion, the program involves the modified Newton-Raphson iteration method in various time steps. Thus, the program is able to predict the position of the parts, starting from the initial position or the conditions of the previous time step.

For the motion study, we went through the following steps:

○ Simplified generation of the components;
○ Assembly of components (see Figure 3, where the components were identified with the same position numbers as in Figure 1);
○ Setting the value of 1000 frames per second while simulating, in order to obtain a high precision simulation;
○ As advanced Motion Analysis settings, the following options were chosen: the GSTIFF type integrator, 25 maximum iterations, 0.0001 initial integrator step size, 0.0000001 minimum integrator step size, 0.01 maximum integrator step size, Jacobian re-evaluation on every iteration;
○ Applying one rotary motor to the rotation axis of each of the eccentric bodies and setting the constant speed (1000 min$^{-1}$), as well as the opposite directions of rotation;
○ Specification of gravity value (9806.65 mm/s$^2$) and direction of action;
○ Particularization of the mates.

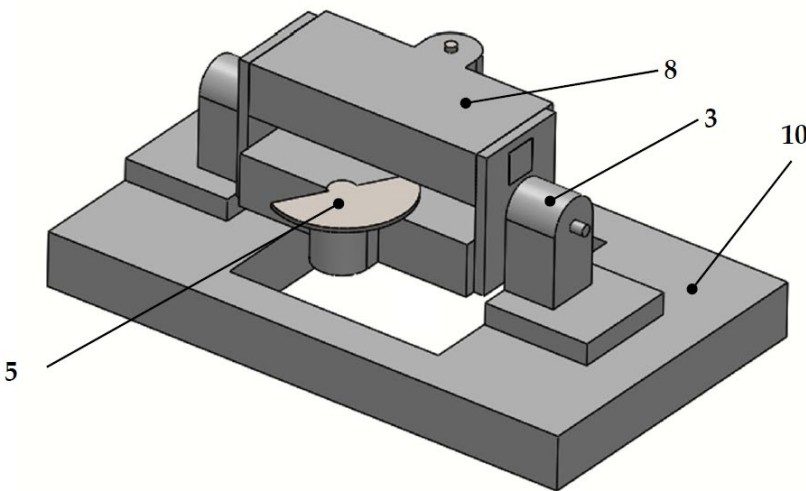

**Figure 3.** Simplified model of the vehicle.

The bottom plate of the assembly (10) was placed on a base plate between the two surfaces, considering that there is no friction. This assumption was made because we aimed to compare the simulation results with the analytical outcomes, where, in the mathematical model, the friction was neglected. The friction is conditioning the contacts between the rigid bodies, but, as we are dealing with rolling friction, this may be neglected in conformity with other comparable mathematical models [3].

As one can observe, the model was constructed in such a way that, for further study, it allows an additional rotation of the housing 8 around the axis of the bearing blocks 3. Furthermore, the analysis time of the study was set to 2 s.

## 5. Results and Discussion

Using the capabilities of the MA module from SW, it was found out that the platform is capable of moving in a certain direction only if the driving masses (*m*) are positioned at the end of the roads of radius *r*, so that in the starting position they are perpendicular to this direction of movement. The following parameters of the IPS were simulated:

### 5.1. Velocity of the Vehicle

Figure 4 shows the velocity evolution of the platform within the 2 s of motion analysis performed at a rotational speed *n* = 1000 rpm of the two eccentric bodies.

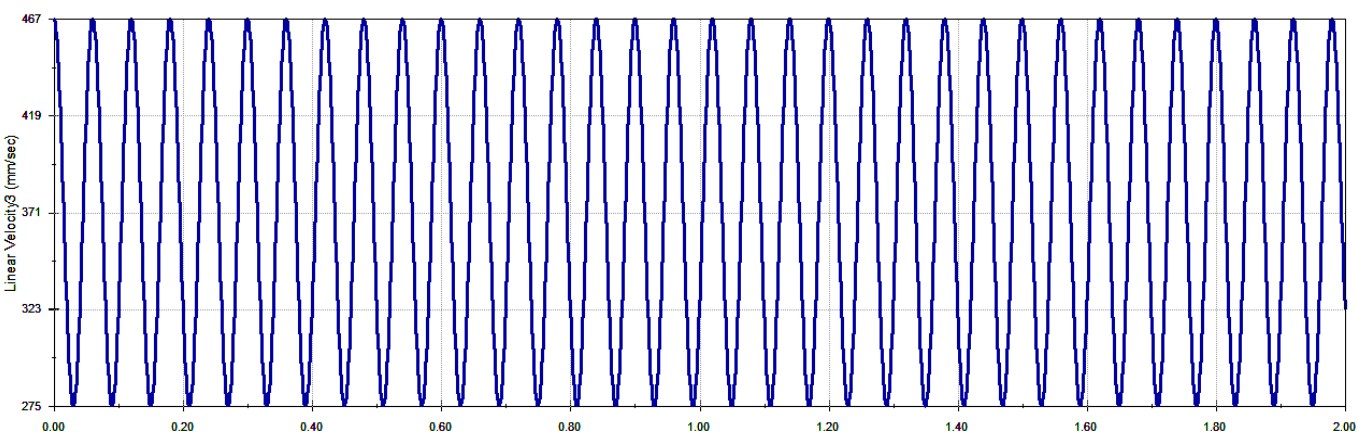

**Figure 4.** Velocity of the vehicle.

As one can observe, the platform's velocity has indeed, as resulted from the analytical model, a constant component and a harmonic one. The frequency with which the velocity

varies is equal to the rotational frequency of the driving masses. The average speed developed by the IPS is $v_{med}$ = 371 mm/s, while the maximum speed at start ($t$ = 0) is 467 mm/s.

Conversely, applying Equation (18) for the calculus of the starting speed of the vehicle, knowing the mass of the platform ($M$ = 1.65 kg), the rotational speed of the two eccentric bodies ($n$ = 1000 rpm), characterized by a mass $m$ = 0.18 kg and a CM radius $r$ = 24.73 mm, we obtained $v_{CM,0}$ = 463 mm/s. This indicates good compliance between the simulation and the analytical model.

*5.2. Power Consumption*

A major benefit of the MA module from SolidWorks is the fact that information regarding the power consumption for driving the IPS is obtained very easily, compared to the analytical approach. Relevant information on this topic is depicted in Figure 5 within the 2 s of motion analysis for driving the platform.

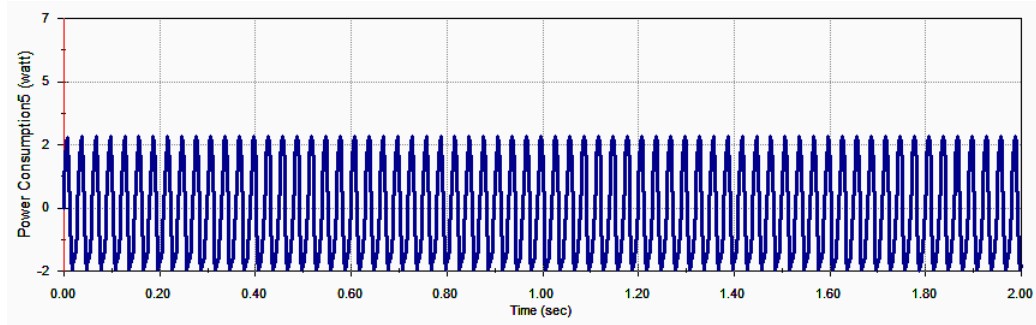

**Figure 5.** Power consumption for driving the IPS.

The theoretical background on which the computation of the power consumption in the MA module is done, starts from the well-known relation of the output power of a motor, which is the product of the torque (T) that the motor generates and the angular velocity ($\omega$) of its output shaft. Furthermore, Newton's Second Law for rotating bodies is applied, so the torque generated by the motor is equal to the product of the mass moments of inertia (I) of the body that is rotated around the axis of the motor and the angular acceleration ($\alpha$) of this body.

As mentioned above, Figure 5 shows the evolution of power consumption. A harmonic variation of the investigated element is also observed in this case. The consumed power varies between a maximum positive value of about 2 W and a negative minimum of −2 W.

Starting from the earlier statements for rotating bodies, the torque consumed by the motor for driving these bodies is equal to the product of the mass moments of inertia (I) of the body, which is rotated around the axis of the motor, and the angular acceleration ($\alpha$) of this body. In addition, the consumed power is equal to the torque and the rotating speed. Therefore, the power consumption for rotating the driving bodies is directly proportional to their angular acceleration. Because of their eccentric shape, the driving bodies will move during a complete rotation, either with positive (accelerated) or negative (decelerated) angular velocity. This means that, during a complete rotation of the driving masses, there is a portion of the circular trajectory in which energy is consumed and another portion in which energy is restored to the system.

The frequency with which the power consumption varies is twice the rotation frequency of the driving masses.

**6. Experimental Proof of the Concept**

Following the simulation outcomes, the authors built a model of the Inertial Propulsion System. For this purpose, some of the simulation data were used to manufacture most of the parts by 3D printing from PLA filament, except for the shafts, bearings, and driving masses.

The experimental tests on the IPS were carried out on a flat surface, on which a starting line was marked with colored adhesive tape. Parallel to this, other lines were drawn with

the same tape at distances of 300, 400, 500, 600, 700, and 1000 mm from the starting line. The thickness of the adhesive strip was 20 μm in order not to influence the speed of the system when passing the strips. Time measurement was conducted with a high-speed camera at 1000 Fps.

The experiments aimed to determine the average speed of movement of the equipment under the following conditions:

- Traveling different distances and comparing the experimental data with the numerical simulation results;
- The variation of the drive speed of the eccentric bodies and the determination of its influence on the average speed of the vehicle;
- The variation of the mass of the eccentrics and the determination of its influence on the average speed of the vehicle.

The speed was determined indirectly by measuring the time required to cover a known distance. For each distance, three measurements of time were conducted, with the average speed being computed as the ratio between the traveled distance and the average travel time.

As mentioned before, the first tests aimed to measure the displacement speed of the IPS and compare it with the average speed obtained by simulation. Therefore, by driving the eccentric bodies with a 1000 rot/min speed, the device had to cover five different distances (300, 400, 500, 600, and 700 mm), with the necessary displacement time being recorded. Table 1 shows the experimental results, the computed average speed, the simulated average speed, and the percentage differences.

**Table 1.** Measurement of the average speed of the IPS.

| Distance [mm] | Measured Time [s] | | | Average Time $t_m$ [s] | Calculated Average Speed $v_m$ [mm/s] | Simulated Average Speed $v_s$ [mm/s] | Percentage Differences [%] |
|---|---|---|---|---|---|---|---|
| | t | t | t | | | | |
| 300 | 0.9 | 0.9 | 0.8 | 0.87 | 344.83 | | 7.05 |
| 400 | 1.1 | 1.1 | 1.2 | 1.13 | 353.98 | | 4.57 |
| 500 | 1.5 | 1.3 | 1.4 | 1.30 | 384.62 | 371 | 3.67 |
| 600 | 1.7 | 1.7 | 1.6 | 1.67 | 359.28 | | 3.15 |
| 700 | 1.9 | 1.9 | 2.0 | 1.97 | 355.33 | | 4.22 |

As one can observe, the differences between the measured speed of the IPS and the simulated results are under 8%. This confirms both that the test procedure was correctly conducted as well as the correctness of the model and the simulation performed. Furthermore, the highest differences were noticed when traveling the shortest distance, where the time measurement was affected by higher errors.

Additionally, the influence of the driving speed of the eccentrics on the vehicle's speed was investigated. For this purpose, the speed of the driving masses was set at four different values (500, 750, 1000, and 1500 rot/min), and the time for traveling 1000 mm was measured three times for each speed of the eccentric masses. Table 2 presents the time measurement results and the calculated average speed.

**Table 2.** Average speed of the IPS at different speeds of the driving masses.

| Speed of the Driving Masses [rot/min] | Distance [mm] | Measured Time [s] | | | Average Time $t_m$ [s] | Calculated Average Speed $v_m$ [mm/s] |
|---|---|---|---|---|---|---|
| | | t | t | t | | |
| 500 | | 10.7 | 10.7 | 10.6 | 10.67 | 93.72 |
| 750 | 1000 | 4.7 | 4.9 | 4.8 | 4.80 | 208.33 |
| 1000 | | 2.7 | 2.7 | 2.6 | 2.67 | 374.53 |
| 1250 | | 1.6 | 1.7 | 1.7 | 1.67 | 598.80 |

The speed of the platform grows with the increase in the driving mass's speed. For instance, by doubling the speed of the driving masses (from 500 to 1000 rpm), the average speed of the platform increases 3.99 times (from 93.72 to 374.53 mm/s). Thus, Figure 6

provides a graphical representation of the correlation between the speed of the driving masses and the average speed of the platform.

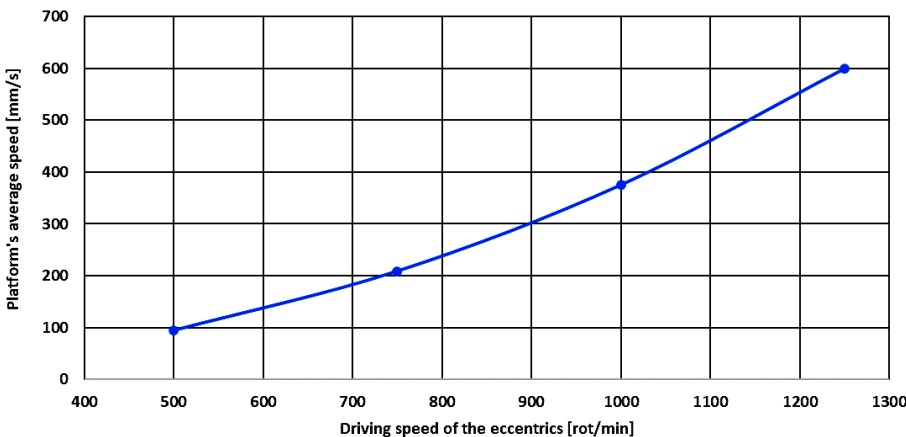

**Figure 6.** Influence of the eccentric's driving speed on the platform's displacement speed.

Finally, the experiments investigated the influence of the eccentric's mass on the speed of the IPS. For this purpose, each of the two driving masses was laser cut from steel sheets (S 235 JR, according to EN 10025-2: 2019 [21]) having higher thicknesses (t = 5, 6, 8, and 10 mm) than the parts used until this stage of the experiments (t = 4 mm).

Each of the supplementary driving masses shown in Figure 7 was incorporated, pair by pair, in the construction of the Inertial Propulsion System and the times for covering a distance of 1000 mm by rotating the driving masses at 500 rot/min were measured. As before, three measurements of time were performed for each pair of driving masses, and the average traveling time was established.

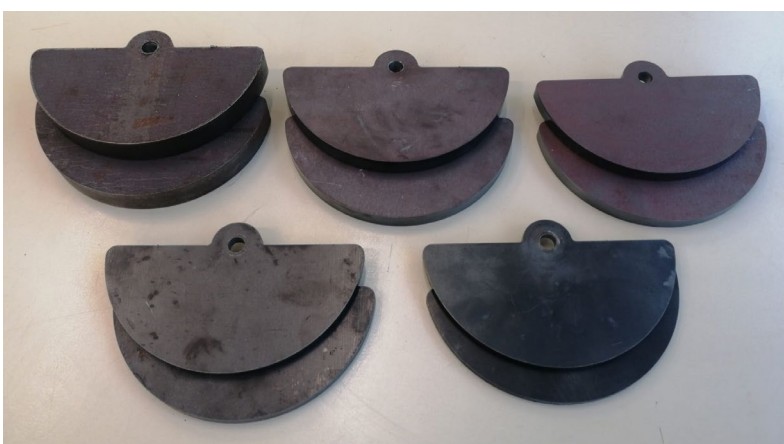

**Figure 7.** Eccentric bodies with different masses (cut from steel sheets with different thicknesses).

Table 3 presents the measurement results and the calculated average displacement speed.

**Table 3.** Average speed of the IPS at different masses of the eccentric bodies.

| Mass of the Eccentric Bodies m [kg] | Distance [mm] | Measured Time [s] | | | Average Time $t_m$ [s] | Calculated Average Speed $v_m$ [mm/s] |
|---|---|---|---|---|---|---|
| | | t | t | t | | |
| 0.180 | | 10.7 | 10.7 | 10.6 | 10.67 | 93.72 |
| 0.225 | | 8.7 | 8.6 | 8.6 | 8.67 | 115.34 |
| 0.270 | 1000 | 7.4 | 7.3 | 7.3 | 7.33 | 136.45 |
| 0.360 | | 5.6 | 5.6 | 5.7 | 5.66 | 176.67 |
| 0.450 | | 4.7 | 4.6 | 4.6 | 4.67 | 214.13 |

The platform's displacement speed increases with the mass of the eccentric bodies. Thus, by multiplying 2.5 times the weight of each driving mass (from 0.18 to 0.45 kg), the speed of the platform increases 2.28 times (from 93.72 to 214.13 mm/s). For a clearer image of how the driving masses influence the traveling speed of the platform, Figure 8 provides a diagram of this interconnection.

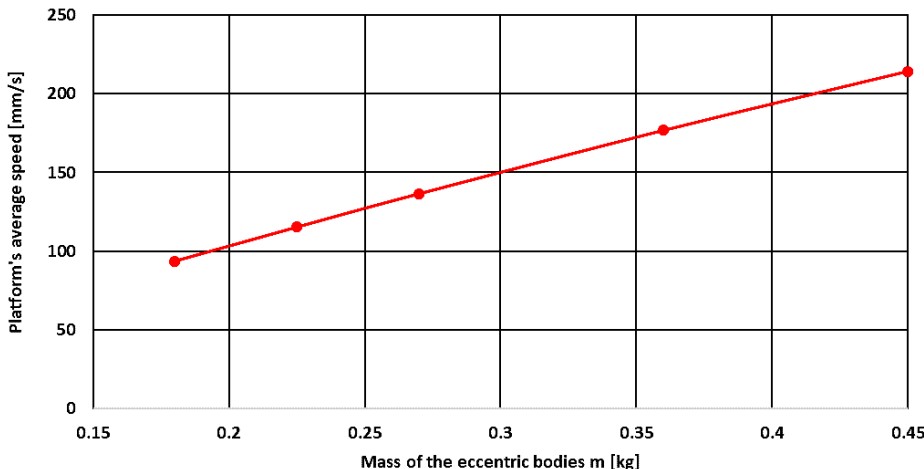

**Figure 8.** Influence of the eccentric's mass on the platform's displacement speed.

The movement speed of the platform does not vary directly proportionally to the mass of the eccentrics because, as Equations (18) and (19) show, in the speed calculation, the constant mass M of the platform is included at the denominator.

## 7. Conclusions

In this paper, an original concept for the propulsion of a mobile platform is presented. The system uses the inertial force generated by two eccentric masses, which are rotating in opposite directions. Verification of the analytically derived kinematic parameters and validation of the concept were performed in SolidWorks by means of motion simulation.

Experimental tests were also performed to demonstrate the ability of the system to produce linear movement. By operating the system over five different distances, it was observed that the differences between the measured average speed of the IPS and the simulated results are under 8%.

Moreover, the influence of the eccentric's driving speed on the vehicle's speed was investigated. The speed of the platform can be increased by increasing the driving mass's speed. That is, by doubling the speed of the driving masses (from 500 to 1000 rpm), the average speed of the platform grows 3.99 times. This result confirms the dependence between these two parameters deduced by analytical means.

Additionally, the influence of the driving body's mass on the speed of the IPS was examined. Incorporating four pairs of eccentric bodies with increasingly higher masses, the platform's average speed was established. It was noticed that by multiplying the weight of each driving mass by 2.5 times (from 0.18 to 0.45 kg), the speed of the platform increases 2.28 times (from 93.72 to 214.13 mm/s). The non-linear dependence between these two parameters was explained by the previously deduced analytical relation.

Finally, the present study confirms that the IPS developed by the authors is functional and capable of generating unidirectional linear movement, thus presenting a possible propulsion system for driving boats or submarines. In this case, the major advantage of the developed system, compared to the current propulsion systems of ships and submarines, is that the propellers and their sealing systems are eliminated.

In addition, the proposed system is the only alternative to running vehicles on very slippery surfaces (mud or ice), where no tire provides the grip necessary to travel in such conditions.

Future research is being planned that will focus on aspects related to the dynamic behavior of the system, with an emphasis on displacement speed uniformity and measures to improve efficiency and power consumption.

**Author Contributions:** Conceptualization, Z.-I.K. and A.G.; methodology, Z.-I.K.; software, C.-O.M. and V.C.; validation, C.-O.M. and Z.-I.K.; formal analysis, S.T.; investigation, S.T.; resources, V.C.; data curation, C.-O.M.; writing—original draft preparation, S.T.; writing—review and editing, A.G.; visualization, A.G.; supervision, Z.-I.K.; project administration, Z.-I.K. All authors have read and agreed to the published version of the manuscript.

**Funding:** This research received no external funding.

**Institutional Review Board Statement:** Not applicable.

**Informed Consent Statement:** Not applicable.

**Conflicts of Interest:** The authors declare no conflict of interest.

## Abbreviations

| | |
|---|---|
| 3D | Three Dimensional |
| CM | Center of Mass |
| FDM | Fused Deposition Modeling |
| Fps | Frames per second |
| IPS | Inertial Propulsion System |
| MA | Motion Analysis |
| PLA | Polylactic Acid |
| SW | SolidWorks |

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
