# Peer review of "Inertial Propulsion of a Mobile Platform Driven by Two Eccentric Bodies"

_applsci, doi:10.3390/app13179511_

Round 1

Reviewer 1 Report

The study is an actual topic for both science world and industry. It is within the scope of the journal. However, there are some major correction parts. These are as follows;

Ø  As described in the introduction, many similar systems have been developed using the same basic physical principle. In other words, the developed system can be evaluated within the scope of another similar product development using a known method. In this respect, this study is more suitable for the subject of Utility Model, considering it in terms of intellectual property rights. As a research paper, it is weak in terms of originality and needs improvement.

 Ø  There are sub-systems such as shaft-bearing, gear and eccentric mechanism on the kinematic chain of the developed system and they consist of many elements. There are mechanical losses in these elements and the system efficiency is highly affected by these losses. This will greatly affect all the obtained results. In this respect, system losses must be taken into account in order to obtain healthy results in such a system.

Ø  It is mentioned that the developed prototype system has been tested. However, sufficient information is not given about the test setup, the equipment used and the measuring instruments. The same inadequacy exists in motion analysis using Solid Works. Please provide detailed information on this subject under the heading of materials and methods.

Ø  In the results section, it is stated that the developed IPS is suitable for driving boats and submarines. However, this system has not been compared with existing propeller systems and its advantages have not been mentioned. Please indicate the advantages of the developed system over existing systems.

Ø  The paper is not free from grammatical errors and it must be reviewed by an English native 

The paper is not free from grammatical errors and it must be reviewed by an English native 

Author Response

Please find attached our Answer to the Review Report of Reviewer 1

Reviewer 2 Report

1.       In contrast to motor propulsion system, what are the advantages for the inertial force propulsion system? It is suggested to add some industrial application examples about the inertial force propulsion system in Introduction by citing some references.

2.       Mobile parts, such as tires, are not seen in Figure 1. How do the developed physical platform move along a straight line on land? Please provide full photos of experimental equipment in section 6, so as to understand how the necessary displacement time is recorded in the experiment.

3.       In the section of experimental proof of the concept, the measured and simulated results in Table 1 are the average speeds, which are not accurate enough. Based on the existing sensor measurement technology and the simulation technology in SolidWorks, it is relatively easy to realize the dynamic real-time measurement of the velocity.

4.       Simulated speed results are missing in Tables 2 and Table 3.

5.       How is the negative power generated in the section of 5.2? How to consider the power consumption of gear friction and wear (shown in figure 1) in the section of 5.2?

 The grammar in the paper needs to be carefully checked.

Author Response

Please find attached our Answer to the Report of Reviewer 2.

Round 2

Reviewer 1 Report

Necessary improvements and corrections have been made. I wish you success in your work

Reviewer 2 Report

The authors have modified their manuscript according to the comments from reviewers. In my opinion, the current version is acceptable.